# Genome-Wide Identification, Characterization, and Expression Analysis of Tubby-like Protein (TLP) Gene Family Members in Woodland Strawberry (*Fragaria vesca*)

**DOI:** 10.3390/ijms231911961

**Published:** 2022-10-08

**Authors:** Shuangtao Li, Guixia Wang, Linlin Chang, Rui Sun, Ruishuang Wu, Chuanfei Zhong, Yongshun Gao, Hongli Zhang, Lingzhi Wei, Yongqing Wei, Yuntao Zhang, Jing Dong, Jian Sun

**Affiliations:** 1Institute of Forestry and Pomology, Beijing Academy of Agriculture and Forestry Sciences, Beijing 100093, China; 2Beijing Engineering Research Center for Strawberry, Beijing 100093, China; 3Key Laboratory of Biology and Genetic Improvement of Horticultural Crops (North China), Ministry of Agriculture, Beijing 100093, China

**Keywords:** tubby-like proteins, *Fragaria vesca*, abiotic stress, evolution pattern, expression analysis

## Abstract

Tubby-like proteins (TLPs) play important roles in plant growth and development and in responses to abiotic stress. However, TLPs in strawberry remain poorly studied. In this study, eight *TLP*s were identified in woodland strawberry (*Fragaria vesca* subspecies *vesca* ‘Ruegen’). Protein structure analysis revealed that the structure of FvTLPs is highly conserved, but evolutionary and gene structure analyses revealed that the evolutionary pattern of *FvTLP* family members differs from that of their orthologous genes in *Arabidopsis*, poplar, and apple. Subcellular localization assays revealed that FvTLPs were localized to the nucleus and plasma membrane. FvTLPs showed no transcriptional activity. Yeast two-hybrid assays revealed that FvTLPs interact with specific FvSKP1s. The expression patterns of *FvTLP*s in different tissues and under various abiotic stresses (salt, drought, cold, and heat) and hormone treatments (ABA (abscisic acid) and MeJA (methyl jasmonate)) were determined. The expression patterns of *FvTLP*s indicated that they play a role in regulating growth and development and responses to abiotic stress in *F. vesca*. The GUS (beta-glucuronidase) activity of *FvTLP1*pro::GUS plants in GUS activity assays increased under salt and drought stress and abscisic acid treatment. The results of this study provide new insights into the molecular mechanisms underlying the functions of TLPs.

## 1. Introduction

Tubby-like proteins (TLPs) are ubiquitous in eukaryotes. TLPs are named for their conserved C-terminal tubby domain, which has a unique helix-filled barrel structure [1], and most plant TLPs also carry a conserved N-terminal F-box domain. TLPs have multiple biological roles in plants. Aside from functioning as transcription factors, they are important components of the SCF (SKP1-cullin-1-F-box) complex [2,3]. TLPs have been reported that respond to biotic stress [4,5,6,7]. OsTLP2 might activate disease resistance pathways by regulating the expression of *OsWRKY13* [8]. In annual *Medicago* species, barley, maize, cassava, apple, cotton, soybean, wheat, and *Salvia miltiorrhiza*, the expression of *TLP*s is affected by various abiotic stresses and hormone treatments [9,10,11,12,13,14,15,16,17]. *TLP1*, *TLP3*, and *TLP5* were identified as potential candidate genes for functional studies in cotton because of their putative functions in drought and salt stress responses [18]. In apple, MdTLP7 enhances abiotic stress tolerance, and the tubby domain plays a key role in abiotic stress tolerance [19]. In chickpea, CaTLP1 positively regulates abiotic stress tolerance and enhances abscisic acid (ABA)-mediated stomatal closure by interacting with protein kinases [20,21]. SlTLFP8 enhances tomato water-deficient resistance by modulating stomatal density [22]. SlTLPs may play a role in fruit ripening, and SlTLP1 and SlTLP2 may participate in ethylene-dependent fruit ripening [23]. CsTLP8, AtTLP3, and AtTLP9 function in ABA- and osmotic stress-mediated seed germination [24,25]. In cotton, GhTULP34 negatively regulates osmotic stress tolerance [26]. AtTLP11 might affect seed development by modulating *NHL6* expression [27], and AtTLP2 regulates the biosynthesis of homogalacturonan, the major polysaccharides constituent of the *Arabidopsis* seed coat mucilage, by activating UDP-glucose 4-epimerase 1 (UGE1) [28]. GmTLP8 confers abiotic stress tolerance in plants via stress-responsive genes [15]. ScTLP12 might contribute to the rolling of leaves in rye [29]. However, few studies were performed on TLPs in strawberry, which is a well know berry sensitive to both biotic stress and abiotic stress.

In this study, we identified *TLP* gene family members in woodland strawberry (*Fragaria vesca*). Gene and protein structure and phylogenetic relationships were analyzed to clarify the evolutionary pattern of *TLP* family members in *F. vesca*. Subcellular localization and transcriptional activity of FvTLPs and protein-protein interactions between FvTLPs and FvSKP1s were analyzed to explore patterns of functional divergence of the TLP family in *F. vesca*. The *cis*-elements (*cis*-acting regulatory DNA elements) and promoters of *FvTLP*s and the expression of *FvTLP*s in different tissues and under various treatments were analyzed to determine the functions of *FvTLP*s. The beta-glucuronidase (GUS) activity of *FvTLP1*pro::GUS plants was analyzed to verify the expression of *FvTLP1*. The results of this study enhance our understanding of the functional characteristics of plant *TLP*s and will aid future studies of the molecular mechanism underlying the functions of TLPs.

## 2. Results

### 2.1. Identification of TLP Gene Family Members in F. vesca

The amino acid sequences of AtTLPs and OsTLPs were used to identify TLPs in the whole genome of *F. vesca*. A total of eight *TLP*s were identified in *Fragaria vesca*, and these were named *FvTLP1* to *FvTLP8* according to their gene IDs. In addition, *FvTLP6* and *FvTLP8* had two and three spliced transcripts, respectively. PCR amplification revealed that *FvTLP8* only has one transcript in ‘Ruegen’. Compared with the coding sequence in the reference genome, the coding sequence of *FvTLP2* in ‘Ruegen’ has an 81-bp deletion and two 1bp mutations (Appendix A). Finally, we identified eight *TLP* genes in ‘Ruegen’ and named them *FvTLP1*-*FvTLP8* according to their gene IDs in the GDR database (Genome Database for Rosaceae, https://www.rosaceae.org, accessed on 6 October 2022), and the two transcripts of *FvTLP6* were named *FvTLP6a* and *FvTLP6b* (Table 1). The length of the *FvTLP*-coding sequences ranged from 1077 to 1308 bp, and the length of FvTLP peptides ranged from 358 to 429 aa. The PI (Predicted Isoelectric) of FvTLPs ranged from 9.11 to 9.78, and the MW (Molecular Weight) ranged from 40.1 to 48.9 kD.

### 2.2. Phylogenetic and Gene Structure Analysis

To clarify the evolutionary relationships among *TLP* genes, a phylogenetic tree of *TLP*s from *Arabidopsis*, rice, apple, poplar, and *F. vesca* was constructed. The tree was divided into three phylogenetic subfamilies: A, B, and C; subfamily A could be further divided into subgroups A1, A2, and A3. All *FvTLP*s belonged to subfamily A, with the exception of *FvTLP5* and *FvTLP8*, which belonged to subfamily B and subfamily C, respectively (Figure 1). The phylogenetic analysis revealed that the TLP proteins of apple and *F. vesca* showed higher homology compared with those of other species. Conserved domain analysis revealed that FvTLP8 only possesses a tubby domain, and all other FvTLPs possess both the F-box and tubby domain (Table 1 and Figure 1). Amino acid sequence analysis of FvTLPs showed that all FvTLPs possess a conserved membrane-localized site, a phosphatidylinositol 4, 5-bisphosphate (PIP2)-binding site (Appendix A) [30]. We speculated that FvTLPs might be located in the plasma membrane. The study of structural divergence can contribute to our understanding of protein function and evolution [31]. To investigate the diversity of *TLP* genes in *F. vesca*, the conserved motifs of FvTLP proteins and the structure of *FvTLP* genes were analyzed (Figure 2). A total of five conserved motifs were identified in FvTLPs (Appendix A); the conserved motifs 1, 2, 4, and 5 were Tub family domains, and motif 3 was the F-box domain. FvTLP1–FvTLP7 possess all identified motifs, and FvTLP8 possesses motifs 1 and 4. This finding is consistent with the results of the conserved domain analysis. Analysis of gene structure showed that *FvTLP8* consisted of nine exons; *FvTLP3*, *FvTLP6a*, and *FvTLP6b*, members of subfamily A2, consisted of five exons; and the other *FvTLP*s consisted of four exons.

### 2.3. Cis-Element and Promoter Analysis

Promoters are important molecular tools for functional gene analysis [32] and for exploring the transcriptional regulation of *FvTLP*s; we identified the *cis*-elements in putative promoters (Figure 3). *Cis*-elements related to the light response, development, hormonal responses, and stress responses were unevenly distributed in the promoters of *FvTLP*s (Figure and Table). The AAGAA-motif and as-1 element are related to development [33]; the AAGAA-motif was detected in all *FvTLP* promoters, with the exception of the *FvTLP5* promoter, and the as-1 element was detected in all *FvTLP* promoters, with the exception of the *FvTLP7* promoter. The RY element is involved in seed-specific regulation and was observed in the *FvTLP1*, *2*, and *4* promoters. E2Fb and MSA-like elements are involved in cell cycle regulation and were only detected in the *FvTLP3* promoter. The GCN4_motif is involved in endosperm expression and was present in the *FvTLP2* and *FvTLP6* promoters. The CAT-box is involved in meristem expression and was only detected in the *FvTLP8* promoter. The O2 site is involved in zein metabolism regulation and was observed in the *FvTLP5*, *6*, and *8* promoters. The hormonal-responsive elements in the promoters of *FvTLP*s included gibberellin-responsive elements (P-box, TATC-box, CARE, GARE-motif, and ERE) [34], salicylic acid element (TCA-element), ABA-responsive element (ABRE), auxin-responsive elements (TGA-element, TGA-box, and AuxRR-core), and MeJA-responsive elements (CGTCA-motif and TGACG-motif). Gibberellin-responsive elements were observed in all *FvTLP* promoters, and auxin-responsive elements and MeJA-responsive elements were detected in all *FvTLP* promoters, with the exception of the *FvTLP7* promoter. ABREs were detected in all *FvTLP* promoters, with the exception of the *FvTLP6* and *FvTLP7* promoters; TCA elements were observed in the *FvTLP1,2*,*5*, and *7* promoters. The stress-responsive elements in the promoters of *FvTLP*s included ARE (anaerobic induction), TC-rich repeats (defense and stress responsiveness), STRE (stress responsiveness), WRE3 (wound-responsive), LTR (low-temperature responsiveness), GC-motif (anoxic specific inducibility), MBS (drought stress responsiveness), WUN-motif (wound-responsive), and DRE core (drought stress responsiveness) elements. The occurrence of other stress-related elements in the promoters of *FvTLP*s was uneven, with the exception of the WRE3 element, which was present in all *FvTLP* promoters. Abundant MYC and MYB-binding motifs were observed in the promoters of *FvTLP*s. 

### 2.4. Subcellular Localization and Transcriptional Activity of FvTLPs

Detecting the subcellular localization of FvTLPs is essential for clarifying their functions. To characterize the subcellular localization of FvTLPs, GFP-tagged FvTLPs were expressed in tobacco leaves. The green fluorescence was observed in the nucleus and plasma membrane through confocal microscopy (Figure 4). This demonstrated that all FvTLPs were localized to the nucleus and plasma membrane.

Yeast one-hybrid assays were performed to analyze the transcriptional activity of FvTLPs. Yeast cells transformed with the positive control grew well on SD/-Trp-His medium and turned blue in the presence of x-α-Gal, whereas yeast cells with the negative control and FvTLPs did not grow on this medium (Figure 5). This suggested that all FvTLPs have no transcriptional activation function in yeast.

### 2.5. FvTLPs Interact with FvSKP1s

Given that the F-box domain is present in the NH_2_-terminal of FvTLPs, we investigated the interaction between FvTLPs and FvSKP1s using Y2H assays. Y2H assays showed that FvTLP1 interacted with FvSKP1a, FvSKP1c, and FvSKP1d; FvTLP3 interacted with FvSKP1a and FvSKP1c; FvTLP4, 5, 6a, and 6b interacted with FvSKP1c and FvSKP1d; and FvTLP7 interacted with FvSKP1a and FvSKP1d (Figure 6). These findings indicated that FvTLPs could serve as subunits of SCF-type E3 ligase. 

### 2.6. Expression Patterns of FvTLPs in Different Tissues of F. vesca

To explore the functional divergence of *FvTLPs*, the expression levels of *FvTLPs* in different tissues were determined. *FvTLP1* and *FvTLP4* were highly expressed in old leaves and flowers; *FvTLP2* and *FvTLP7* were highly expressed in the roots; and the expression of *FvTLP3* was lower in the roots than in the other tissues. *FvTLP5*, *FvTLP6a*, and *FvTLP6b* were highly expressed in flowers, and *FvTLP8* was highly expressed in the petiole (Figure 7A). The expression of *FvTLP1, FvTLP6a*, *FvTLP6b*, *FvTLP7*, and *FvTLP8* in achene tissue and the receptacle decreased with fruit ripening. No significant variation was observed in the expression of *FvTLP2* and *FvTLP5* in achene during the fruit ripening process; however, the expression of these genes decreased in the receptacle as fruits ripened. The expression of *FvTLP3* in achene and receptacle increased with fruit ripening. The expression of *FvTLP4* decreased in achene tissue but increased in the receptacle as fruits ripened (Figure 7B,C).

### 2.7. Responses of FvTLPs to Abiotic Stress and Exogenous Hormones in F. vesca Leaves

To explore the potential function of *FvTLPs*, the expression of *FvTLPs* was detected under various abiotic stresses and hormone treatments. Under salt stress, the expression of *FvTLP1*, *FvTLP3*, *FvTLP4*, *FvTLP6a*, *FvTLP7*, and *FvTLP8* was up-regulated, and the expression of *FvTLP2* and *FvTLP5* was down-regulated (Figure 8A). The expression of *FvTLP1*, *FvTLP2*, *FvTLP4*, and *FvTLP6a* was induced by PEG treatment compared with the control (Figure 8B). Under heat stress, the expression of all *FvTLP*s was induced, with the exception of *FvTLP3*, and the expression of *FvTLP1* and *FvTLP4* was markedly induced by heat stress (Figure 8C). Under cold stress, the expression of all *FvTLP*s was reduced, and the expression of *FvTLP1*, *FvTLP6a*, *FvTLP6b*, and *FvTLP8* was significantly inhibited by cold stress. The expression of *FvTLP6b* was initially down-regulated and later significantly up-regulated (Figure 8D). Under ABA treatment, the expression of all *FvTLP*s was down-regulated, with the exception of *FvTLP1* (Figure 8E). Under MeJA treatment, the expression of *FvTLP1*, *FvTLP2*, *FvTLP3*, *FvTLP5*, and *FvTLP6A* was up-regulated, and the expression of *FvTLP4*, *FvTLP7*, and *FvTLP8* was down-regulated (Figure 8F).

### 2.8. FvTLP1 Expression Responds to Abiotic Stress

The expression of *FvTLP1* under various abiotic stresses and hormone treatments was assayed by GUS staining of *Arabidopsis* expressing the GUS reporter gene driven by the *FvTLP1* promoter. The GUS expression of *FvTLP1*pro::GUS plants under NaCl, mannitol, or ABA treatment was significantly higher than that under normal conditions. NaCl, mannitol, and ABA treatment significantly induced GUS expression during seed germination (Figure 9). These results indicated that *FvTLP1* might play a role in the response to abiotic stress and in ABA- and osmotic stress-mediated seed germination.

## 3. Discussion

In this study, we identified eight *TLP*s in woodland strawberry (*F. vesca*). The number of *TLP*s was significantly less in woodland strawberry than in other diploid plants, such as *Arabidopsis* (11), rice (14), poplar (11), maize (15), tomato (11), and soybean (22) [11,15,23,35], and no paired paralogous *TLP* genes were identified in *F. vesca* (Figure 1) as has been observed in *TLP*s in *Arabidopsis*, rice, and poplar [35]. Intron/exon structures provide important insights into the evolutionary relationships among members of gene families [36]. The intron/exon structures of *FvTLP*s significantly differed from those of orthologous genes in *Arabidopsis*, poplar, and apple. All *FvTLP*s contained four or five exons, with the exception of *FvTLP8*, which contained eight exons (Figure 2), and the number of exons in their orthologous genes in *Arabidopsis*, poplar, and apple ranged from two to nine [13,35]. These results indicated that the evolutionary history of *TLP* genes in *F. vesca* differed from that of orthologous genes in other plant species. Although post-transcriptional modifications and variations in the transcripts of a gene are affected by the number of exons, it has been stated that genes with a low number of exons are induced faster in response to external [37,38].

The results of the conserved domain analysis, motif analysis, and amino acid sequence analysis showed that the structure of FvTLPs is conserved (Figure 1, Figure 2 and Appendix A). The subcellular localization of plant TLPs is diverse, as plant TLPs have been reported to be localized to the cell wall, plasma membrane, cytosol, and nucleus [21,22,24]. The PIP2-binding site contributes to the plasma membrane localization of TLPs [30]; all FvTLPs possess a conserved PIP2-binding site in the tubby domain, and all FvTLPs were localized to the plasma membrane (Figure 4). In *F. vesca*, all FvTLPs possess a conserved F-box domain, with the exception of FvTLP8 (Figure 1). F-box proteins responsible for substrate recognition for the SCF ubiquitin ligase complexes play important roles in plant development, light and phytohormone signaling, mitotic processes, and responses to abiotic stress [39]. In this study, we showed that FvTLPs can interact with specific FvSKP1s (Figure 6), and this has also been observed among TLPs in *Arabidopsis*, wheat, and cucumber [3,24,25]. These findings indicate that the functions of FvTLPs are conserved.

TLPs have been reported to play important roles in the response to abiotic stress in chickpea, apple, cotton, tomato, cucumber, and soybean [15,19,20,22,25,26]. *FvTLP*s also play a role in the response to multiple types of stress and hormone treatments. The expression of *FvTLP1* was affected by various abiotic stresses and hormone treatments; *FvTLP1* expression was strongly induced by heat stress, ABA treatment, and MeJA treatment and strongly reduced by cold stress. The expression of *FvTLP2* was induced by heat stress, as well as PEG and MeJA treatment. The expression of *FvTLP3* was strongly induced by salt stress and MeJA treatment. The expression of *FvTLP4* was strongly induced by heat stress. The expression of *FvTLP5* was strongly induced by MeJA treatment. The expression of *FvTLP6a* was strongly induced by PEG treatment and strongly reduced by cold stress. Under cold treatment, the expression of *FvTLP6b* was first significantly decreased and then strongly increased. The expression of *FvTLP8* was significantly reduced by cold stress and MeJA treatment (Figure 8). These results indicated that *FvTLP*s might play a role in responses to abiotic stress.

ABA and JA play indispensable roles in controlling leaf senescence [40,41]. The expression levels of *FvTLP1* and *FvTLP4* were significantly higher in old leaves than in young leaves. The expression of *FvTLP1* was strongly induced by ABA and MeJA, and the expression of *FvTLP4* was strongly reduced by ABA and MeJA (Figure 7 and Figure 8). These findings indicated that *FvTLP1* and *FvTLP4* might play a role in regulating leaf senescence via the ABA and JA signaling pathways. *SlTLP*s have been reported to play a role in ethylene-dependent fruit ripening. The fruit ripening of strawberry is non-climacteric, and ABA and MeJA play a key role in the ripening of strawberry fruit [42,43]. The expression of *FvTLP1*, *FvTLP6b*, and *FvTLP8* in the achene and receptacle significantly decreased with fruit ripening, whereas the expression of *FvTLP3* in the achene and receptacle increased with fruit ripening (Figure 7). The expression of *FvTLPs* was affected by ABA and MeJA treatment, which suggested that *FvTLP*s might play a role in fruit ripening via the ABA and JA signaling pathways. Overall, these results indicated that *FvTLPs* play key roles in regulating the growth and development of strawberries.

Alternative splicing plays a key role in the abiotic stress response by mediating the production of multiple transcripts and increasing proteome complexity; ABA plays a key role in regulating alternative splicing during stress responses [44]. Abiotic stress can also modulate gene expression via alternative splicing pathways [45,46,47,48]. LSM5 (Sm-like protein 5) is involved in the heat shock response by degrading aberrant heat stress-inducible transcripts via mRNA splicing and decapping [49]. HIN1 (HAI1-Interactor 1), a plant-specific RNA-binding protein, promotes growth during drought stress and improves splicing efficiency at intron retention sites [50]. In *Arabidopsis*, *SRAS1* (*salt-responsive alternatively spliced gene 1*) produces two splicing variants: *SRAS1.1* and *SRAS1.2*. *SRAS1.1* and *SRAS1.2* exhibit opposite expression patterns in response to salt stress. Salt stress induces the expression of *SRAS1.1* and reduces the expression of *SRAS1.2*; *SRAS1.1* increases salt tolerance, and *SRAS1.2* decreases salt tolerance. Salt stress-triggered alternative splicing regulates the ratio of SRAS1.1/SRAS1.2, which affects the stability of CSN5A (COP9 signalosome 5A), a core molecular switch between the stress response and development, and regulates the balance between plant development and salt tolerance [51]. In cotton, 93 alternative splicing events have been identified in 29 of 35 *GhTULP* genes, but the function of the alternative splicing events in *GhTULP*s in response to environmental stress and other biological processes remains unclear. In this study, *FvTLP6* generated two splicing variants: *FvTLP6a* and *FvTLP6b*. *FvTLP6a* and *FvTLP6b* presented distinct expression patterns under abiotic stress and hormone treatment and at different ripening stages of fruits (Figure 7 and Figure 8). This indicated that *FvTLP6a* and *FvTLP6b* might play different roles in abiotic stress responses and fruit ripening regulation. 

LTR *cis*-element has been shown to play key roles in the response to low-temperature stress [52]. MdbHLH33 binds to the promoter of *MdCBF2* at the LTR *cis*-acting element to activate the expression of *MdCBF2*, which enhances the cold tolerance of transgenic callus [53]. MdbHLH33 can bind to the LTR *cis*-element of the *MdMYBPA1* promoter to activate the expression of *MdMYBPA1*, which promotes anthocyanin biosynthesis under low-temperature stress [54]. In *Lotus corniculatus*, the salt-responsive transcription factor LcERF056 enhances salt stress tolerance by binding to the GCC box or DRE of reactive oxygen species-related genes [55]. SlDREBA4 specifically binds to the DRE elements of *Hsp* (heat shock proteins) genes in response to high-temperature stress [56]. The MBS motif is responsible for ABA and abiotic stress-induced gene expression [57,58]. Low-temperature stress induces the expression of *MdMYB2*, which binds to the promoter of *MdSIZ1* via the MBS motif and enhances the transcription of MdSIZ1, thereby promoting the sumoylation of MdMYB1. Sumoylated MdMYB1 increases anthocyanin accumulation and cold stress tolerance in apple [59]. In *S. cerevisiae*, the STRE *cis*-element plays a role in the response to various types of stress [60], and YCP4 regulates the freeze-thaw tolerance of yeast by modulating the activation of the STRE-mediated genes *CTT1* and *HSP12* [61]. *Cis*-element and promoter analysis showed that the promoters of *FvTLP*s contain various stress-related *cis*-elements, and *FvTLP*s were expressed in response to various types of abiotic stress (Figure 8). Results of GUS activity assays indicated that *FvTLP1* might play a role in ABA- and osmotic stress-mediated seed germination (Figure 9). AtTLP3 and AtTLP9 play redundant roles in regulating ABA- and osmotic stress-mediated seed germination, and CsTLP8 and GhTULP34 negatively regulate seed germination under osmotic stress. The molecular mechanisms underlying these regulatory processes remain unclear. Our results indicate that further study of promoters might provide new insights into the functions of TLPs.

We systematically identified *TLP* gene family members in woodland strawberry (*F. vesca*). Gene structure, protein structure, and phylogenetic analyses, along with the results of subcellular localization, transcriptional activity, and protein interaction assays, revealed that the structure and function of *TLP* genes are highly conserved despite variation in the evolutionary history of *TLP* genes among plants. Expression patterns of *FvTLP*s in different tissues and under various abiotic stresses, coupled with GUS activity assays in *FvTLP1*pro::GUS plants, revealed that *FvTLP*s play a role in abiotic stress responses in *F. vesca*. Generally, the results of this study provide new insights into the molecular mechanisms of TLPs.

## 4. Materials and Methods

### 4.1. Genome-Scale Identification of FvTLPs

The amino acid sequences of AtTLPs and OsTLPs were downloaded from the TAIR database (http://www.arabidopsis.org/, accessed on 12 December 2020) and RGAP database (http://rice.plantbiology.msu.edu/, accessed on 12 December 2020), respectively. The amino acid sequences of model plants were used as BLAST queries against the woodland strawberry genome (V4.0) to identify TLPs in *Fragaria vesca.* All the candidate sequences were further verified by a hidden Markov model (HMM) search using the Pfam (http://pfam.xfam.org/, accessed on 5 January 2021) and SMART (http://smart.embl-heidelberg.de/, accessed on 5 January 2021) databases. The coding sequences of *FvTLP*s were amplified by PCR from ‘Ruegen’ cDNA. Next, the online ExPASy Proteomics Server (http://www.expasy.org/, accessed on 5 March 2021) was used to analyze the physical and chemical characteristics of TLPs, including the number of amino acids, nucleotide data, molecular weight (MW), and isoelectric point (PI).

### 4.2. Gene Structure, Phylogenetic, and Cis-Element Analysis of FvTLPs

The exon-intron structure of *FvTLP*s was analyzed using TBtools. The conserved motifs of FvTLPs with default parameters were analyzed using MEME (https://meme.nbcr.net/meme/cgi-bin/meme.cgi, accessed on 9 March 2021), and the maximum number of motifs was set as five. The conserved domains of FvTLPs were analyzed using the Pfam database (http://pfam.xfam.org/, accessed on 12 March 2021). The phylogenetic tree was constructed using the MEGA 10.0 program and modified by iTOL (https://itol.embl.de/, accessed on 21 March 2021). We extracted sequences 2.0 kb upstream from the translation start site of *FvTLPs*; *cis*-elements were predicted using the online program PlantCARE (http://bioinformatics.psb.ugent.be/webtools/plantcare/html/, accessed on 26 February 2021) and visualized using TBtools.

### 4.3. Plant Materials and Treatment

Seeds of woodland strawberry (*Fragaria vesca* subspecies *vesca* ‘Ruegen’) were grown in 1-L diameter pots containing Hoagland nutrient solution in a growth chamber (light intensity: approximately 100 mol m^−^^2^ s^−^^1^; photoperiod: 16 h (light)/8 h (dark); temperature: 25 °C/18 °C; relative humidity: 60%. The 2-month-old plants were treated with either 150 mM NaCl, 10% PEG_6000_, 100 μM ABA, 100 μM methyl jasmonate (MeJA), cold (4 °C), or heat (40 °C) for 0 h, 1 h, 3 h, 6 h, 9 h, 12 h, 24 h, and 48 h, and the leaves from five individual plants were collected. 

### 4.4. RNA Isolation and qRT-PCR Analysis

The RNA was extracted from the samples following our previously published protocol [62]. Quantitative real-time PCR was performed on a 7500 Real-time PCR System (Applied Biosystems) with the following thermal cycling conditions: 95 °C for 20 s followed by 40 cycles at 95 °C for 5 s, 60 °C for 10 s, and 72 °C for 20 s. A melting curve (61 cycles at 65 °C for 10 s) was generated to determine the specificity of the reaction. The relative gene expression levels were quantified using the 2^-^^△△^^CT^ method. *FveCHC1* [63] was used as the internal reference gene; the primers of *FvTLP* genes were designed using Primer Premier 7.0 software. The primer pairs are listed in Appendix A.

### 4.5. Subcellular Localization of FvTLPs

The coding sequences of *FvTLP*s were inserted into the pSuper1300 vector [64] to form SUPERpro: FvTLP-GFP. The fusion constructs were transferred into *Agrobacterium tumefaciens* strain GV3101 and then injected into tobacco leaves. The fluorescence signal in tobacco epidermal cells was detected by confocal microscopy (FV1000, Olympus, Tokyo, Japan), and cell nuclei were stained with DAPI (4’,6-diamidino-2-phenylindole). The experiment was performed three times.

### 4.6. Transactivation Activity Assay for FvTLPs in Yeast

The coding sequences of *FvTLP*s were inserted into the pGBKT7 vector. The fusion constructs were transferred into Y2HGold yeast. Transcriptional activity was assayed as described by Gong et al. [65]. CsTLP8 was used as a positive control, and pGBKT7 plasmid was used as a negative control [25]. 

### 4.7. Yeast Two-Hybrid (Y2H) Assays

Y2H assays were performed following the methods of Zhang et al. [66]. Briefly, the coding sequences of *FvSKP1*s were inserted into the pGADT7 vector. The constructs with FvTLPs-BD and FvSKP1s-AD were then co-transformed into AH109 yeast. Protein interactions were screened on DDO medium (SD/-Leu/-Trp) and QDO medium (SD/-Trp/-Leu/-His/-Ade) supplemented with X-α-galactosidase. The co-transformation of pGADT7-T and pGBKT7-53 was used as positive control, and the co-transformation of pGADT7-T and pGBKT7-Lam was used as negative control (Appendix A).

### 4.8. Plasmid Construction and Arabidopsis Transformation

The 1600-bp promoter of *FvTLP1* was inserted into the pCAMBIA1391 vector (Marker Gene Technologies, Inc., Eugene, OR, USA); the fusion construct was transferred into *Agrobacterium tumefaciens* strain GV3101 and then integrated into Columbia wild-type *Arabidopsis* (Col-0) through the floral dipping method. The transgenic lines were identified by resistance to the antibiotic hygromycin B and PCR amplification (Appendix A). T3 transgenic plants were used in subsequent experiments.

### 4.9. Histochemical Assay for GUS Activity

Transgenic *Arabidopsis* seedlings were grown on half-strength MS medium, and 3-week-old seedlings were transferred to half-strength MS medium supplemented with 150 mM NaCl, 200 mM mannitol, or 100 μM ABA for two days. Sterilized seeds were sown on half-strength MS medium supplemented with or without 75 mM NaCl, 75 mM mannitol, or 0.25 μM ABA for five days. After the stress treatment, histochemical GUS staining was performed following the method of Jameel et al. [67]. Briefly, the prepared materials were immersed in GUS staining solution buffer and incubated overnight at 37 °C. The plant tissues were then decolorized with 95% ethanol and observed under a light microscope.

### 4.10. Statistical Analysis

Each experiment was repeated three times. Statistical analyses were conducted using SPSS 20.0 software (IBM Corp. Armonk, NY, USA), and statistical comparisons were performed using a t-test in SPSS.

## Figures and Tables

**Figure 1 ijms-23-11961-f001:**
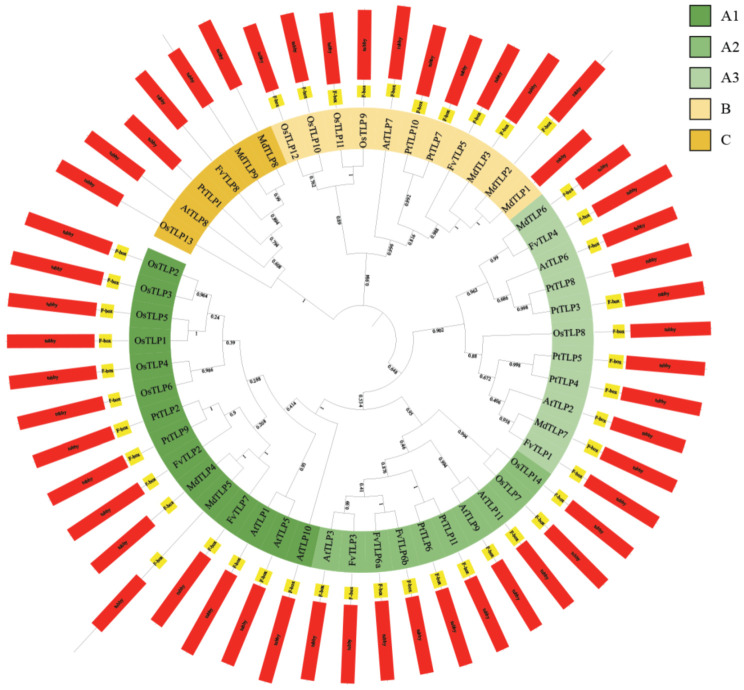
Phylogenetic tree and conserved domains of TLPs. The phylogenetic tree was constructed using the maximum likelihood method based on multiple sequence alignments of TLP proteins from *Arabidopsis thaliana*, *Oryza sativa*, *Malus domestica*, *Populus trichocarpa*, and *Fragaria vesca*. The tree was divided into three phylogenetic subfamilies, which are shown in different colors. Positions of the conserved F-box (yellow rectangle) and tubby (red rectangle) domains are also shown.

**Figure 2 ijms-23-11961-f002:**
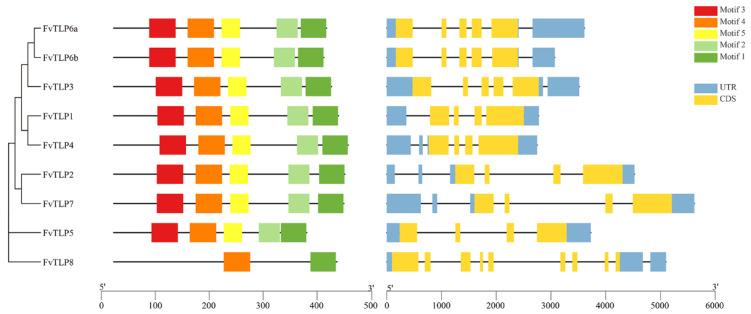
Conserved motif composition of FvTLPs and gene structure of *FvTLP*s. Distribution of conserved motifs in FvTLPs (**left**) and intron–exon structures of *FvTLP* genes (**right**). Rectangles of different colors represent different motifs, blue rectangles represent UTRs, black lines represent introns, and ginger rectangles represent exons.

**Figure 3 ijms-23-11961-f003:**
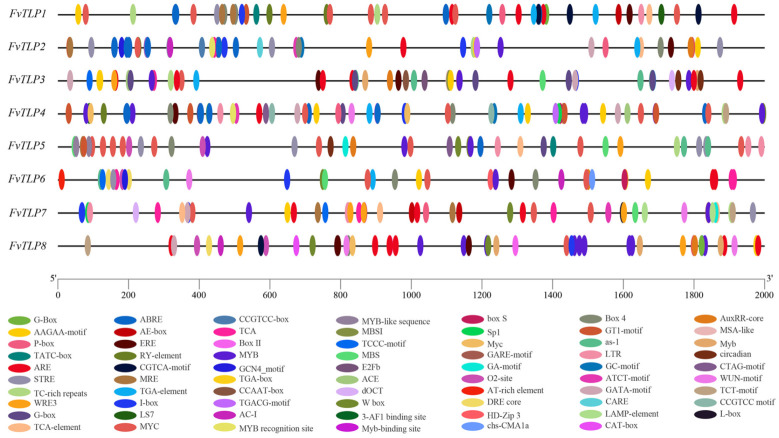
*Cis*-element analysis of *FvTLP* promoters. The distribution of *cis*-elements in the 2000-bp upstream promoter and the different types of *cis*-elements are represented by different colors.

**Figure 4 ijms-23-11961-f004:**
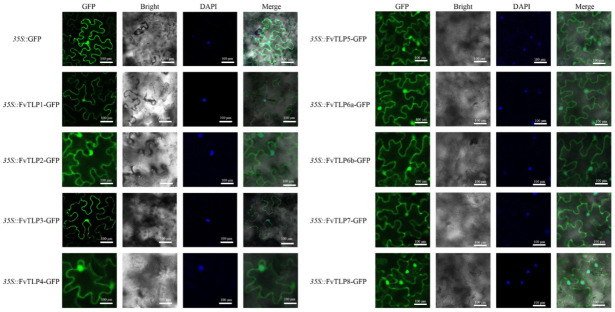
Subcellular localization of FvTLPs. FvTLP-GFP and GFP were transiently expressed in tobacco epidermal cells and then observed with a confocal microscope. Cell nuclei were stained with DAPI.

**Figure 5 ijms-23-11961-f005:**
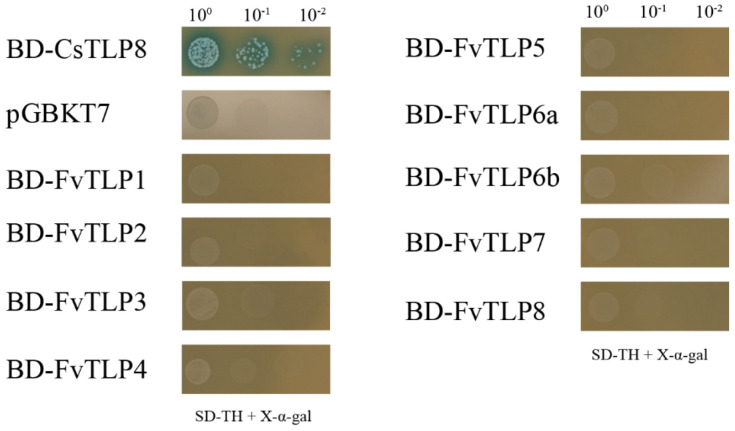
Transcriptional activity analyses of FvTLPs by yeast one-hybrid assays. Yeast cells transformed with the empty vector pGBKT7 (negative control, NC), pGBKT7-*CsTLP8* (positive control, PC), or pGBKT7-*FvTLP*s were streaked on SD/-Trp/-His/X-α-gal medium.

**Figure 6 ijms-23-11961-f006:**
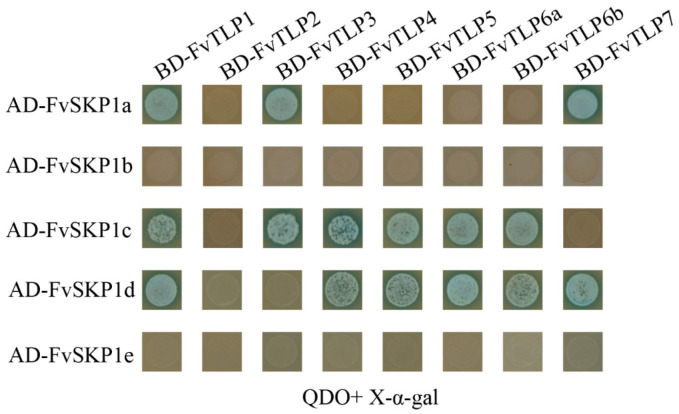
FvTLPs interact with FvSKP1s. Protein–protein interactions analyzed by yeast two-hybrid assays. Yeast cells transformed with BD- *FvTLP*s +AD- *FvSKP1*s were streaked on SD/-Trp/-Leu/-Ade/-His/X-α-gal medium.

**Figure 7 ijms-23-11961-f007:**
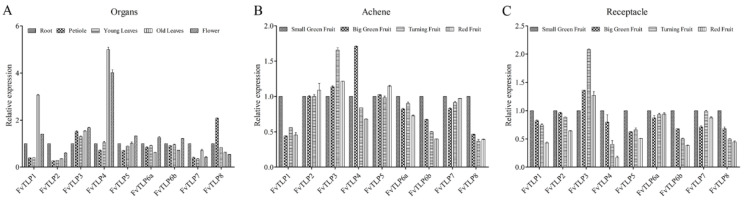
Expression patterns of *TLP* genes in *Fragaria vesca*. (**A**) Expression profiles of *FvTLP*s in different tissues (root, petiole, young leaves, old leaves and flower). Expression profiles of *FvTLP*s in achene (**B**) and receptacle (**C**) at different stages of fruit development (small green, big green, turning, and red).

**Figure 8 ijms-23-11961-f008:**
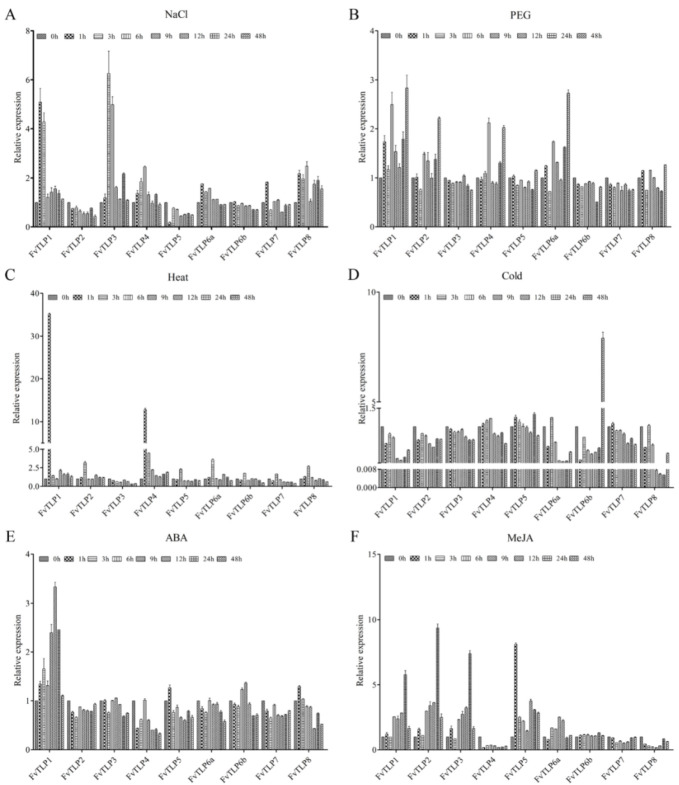
Responses of *FvTLP*s to various abiotic stresses and hormone treatments. The expression profile of *FvTLP*s in the leaves under NaCl (150 mM) treatment (**A**), PEG_6000_ (10%) treatment (**B**), heat (42 °C) treatment (**C**), cold (4 °C) treatment (**D**), ABA (100 μM) treatment (**E**), and MeJA (100 μM) treatment (**F**).

**Figure 9 ijms-23-11961-f009:**
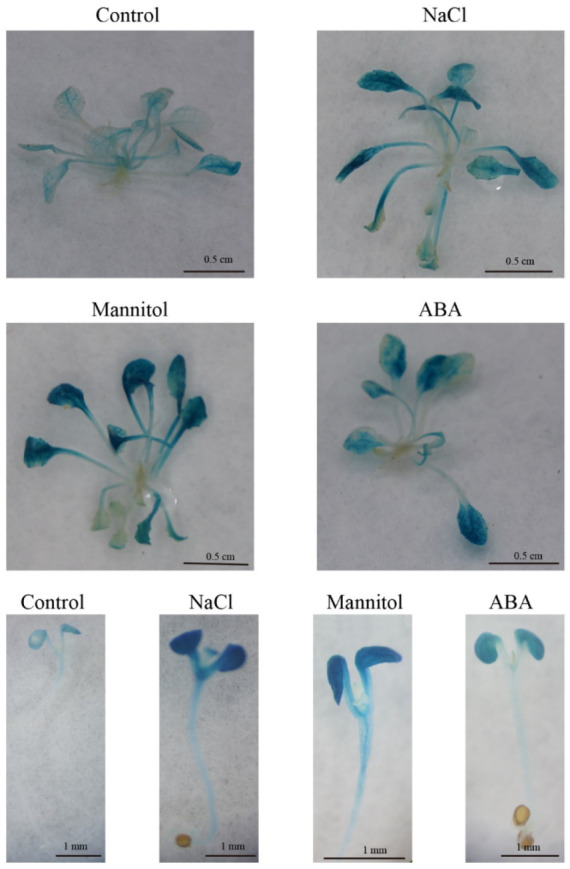
Histochemical GUS staining of *FvTLP*1pro::GUS transgenic *Arabidopsis*. Three-week-old seedlings growing on half-strength MS medium were transferred to half-strength MS medium without or with 150 mM NaCl, 200 mM mannitol, or 100 μM ABA. Seeds of *FvTLP*1pro::GUS transgenic *Arabidopsis* were surface-sterilized and germinated on half-strength MS medium supplemented without or with 75 mM NaCl, 75 mM mannitol, or 0.25 μM ABA.

**Table 1 ijms-23-11961-t001:** Identification and characteristics of *TLP* genes in *Fragaria vesca*.

Gene Name	Gene ID	CDS (bp)	Peptide (aa)	Predicted Isoelectric	Molecular Weight (kD)	Amino Acid Residue Localization of F-Box Domain		Amino Acid Residue Localization of Tubby Domain
*FvTLP1*	FvH4_1g13070	1254	417	9.31	46.9	54–99		117–412
*FvTLP2*	FvH4_2g13720	1290	429	9.74	48.1	53–95		116–424
*FvTLP3*	FvH4_2g24790	1215	404	9.46	44.9	49–94		114–399
*FvTLP4*	FvH4_2g39640	1308	435	9.11	48.9	58–100		125–340
*FvTLP5*	FvH4_3g18250	1077	358	9.43	40.1	41–86		106–353
*FvTLP6a*	FvH4_4g17170.t1	1188	395	9.78	44.2	37–90		102–390
*FvTLP6b*	FvH4_4g17170.t2	1173	390	9.69	43.6	37–90		102–385
*FvTLP7*	FvH4_5g30230	1284	427	9.47	48.0	53–95		116–422
*FvTLP8*	FvH4_6g28190	1245	414	9.39	46.2			166–408


## Data Availability

The data presented in this study are available on request from the corresponding author.

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
