# Peer review of "Genome-Wide Identification, Characterization, and Expression Analysis of Tubby-like Protein (TLP) Gene Family Members in Woodland Strawberry (Fragaria vesca)"

_ijms, 2022, doi:10.3390/ijms231911961_

Round 1
Reviewer 1 Report
The authors have done a good study related to genome analysis of TLP gane family in woodland strawberry and new insights of function the TLPs are revealed. In my opinion, this manuscript has potential for publishing in IJMS. I have some comments, include:
- Abstract and Introduction are well written.
- Please add the Bootstrap value to figure 1.
- I recommend using ML method instead of the NJ for constructing phylogeny tree.
- The sequences or motifs logo of conserved motifs should be provided in supplementary data.
- Gene name should be italicized in figure 3.
- I suggest to add this sentence to line 249: Although post-transcriptional modifications and variations in the transcripts of a gene are affected by the number of exons, it has been stated that genes with low number of exons are induced faster in response to external (Koralewski and Krutovsky, 2017; Heidari et al, 2022).
- Discussion and Materials and Methods also are well provided.
References:
- Koralewski and Krutovsky, 2017: https://doi.org/10.1371/journal.pone.0018055
- Heidari et al, 2022: https://doi.org/10.3390/agronomy12102253
Author Response
Dear Reviewer,
Thank you very much for professional and valuable evaluation and comments. According to your suggestion we reconstructed phylogeny tree using Maximum Likelihood method, and bootstrap value has been added in Figure 1. The motif logo of conserved motifs has been added in Figure S3. According to your reminder gene name in Figure 3 has been italicized. And we added the sentence according to your suggestion in line 278-281, to make the discussion more comprehensive.

Reviewer 2 Report
Genome-wide identification, characterization, and expression analysis of tubby-like protein (TLP) gene family members in woodland strawberry.
The manuscript describes an important study of identification and characterization of Tubby-like protein (TLP) gene family in woodland strawberry (Fragaria Vesca) which is an important relative of cultivated strawberry (Fragaria ananassa).The authors conducted bioinformatic and experimental work to identify 8 TLPs in in the genome of F.vesca, interestingly the phylogenetic and gene structure analysis showed that strawberry FvTLP genes contained conserved Tubby-like and F-box structural domains but also showed variations in intron-exon structures from Arabidopsis, rice and apple TLPs. The promoter analysis conducted by authors identified cis-elements in FvTLPs related to light response, growth and development, hormone responses, stress responses. Authors also showed that FvTLPs were detected in nucleus and plasma membrane of tobacco epidermal cells and no transcriptional activation function of FvTLPs was discovered in yeast cells. The interaction studies by authors showed that these FvTLPs interact with FvSKP1s indicating FvTLPs could serve as subunits of SCF-type E3 ligase. Importantly, the FvTLPs members were shown to be differently expressed in response to various abiotic stresses and hormone treatments indicating that these genes might be important for regulating stresses and growth and development in Fragaria.
Overall, the authors did an important work on identification and characterization of FvTLP gene family. This can be used as foundational work to further explore and understand the detailed functional role of individual member of the FvTLP family. The manuscript is well written and easy to follow. However, the authors are encouraged to re-read the manuscript. In line 207 there is a minor spelling error Under is wrongly written as ‘Unser’.
Author Response
Dear Reviewer
Thank you very much for professional and valuable evaluation and comments. And we are very appreciating your pointing out and revising the English language of the manuscript. We check and revise whole manuscript as your suggestion.
